# A Comparison of Total Food Intake at a Personalised Buffet in People with Obesity, before and 24 Months after Roux-en-Y-Gastric Bypass Surgery

**DOI:** 10.3390/nu13113873

**Published:** 2021-10-29

**Authors:** Natasha Kapoor, Werd al Najim, Camilo Menezes, Ruth K Price, Colm O’Boyle, Zsolt Bodnar, Alan C Spector, Neil G Docherty, Carel W le Roux

**Affiliations:** 1Diabetes Complications Research Centre, School of Medicine, University College Dublin, D04 V1W8 Dublin, Ireland; natashankapoor@gmail.com (N.K.); werd.al-najim@ucd.ie (W.a.N.); camilo.menezes@ucd.ie (C.M.); carel.leroux@ucd.ie (C.W.l.R.); 2The Nutrition Innovation Centre for Food and Health, Ulster University, Coleraine BT52 1SA, UK; rk.price@ulster.ac.uk; 3Department of Surgery, Bon Secours Hospital, T12 DV56 Cork, Ireland; cjoboyle@thecorkclinic.eu; 4Department of Surgery, Letterkenny University Hospital, F92 AE81 Letterkenny, Ireland; drbozsolt@gmail.com; 5Department of Psychology and Program in Neuroscience, Florida State University, Tallahassee, FL 32306, USA; spector@psy.fsu.edu; 6Centre for Diabetes, Ulster University, Coleraine BT52 1SA, UK

**Keywords:** obesity, bariatric surgery, food intake, food preferences, macronutrient intake

## Abstract

Long-term reductions in the quantity of food consumed, and a shift in intake away from energy dense foods have both been implicated in the potent bariatric effects of Roux-en-Y gastric bypass (RYGB) surgery. We hypothesised that relative to pre-operative assessment, a stereotypical shift to lower intake would be observed at a personalised *ad libitum* buffet meal 24 months after RYGB, driven in part by decreased selection of high energy density items. At pre-operative baseline, participants (*n* = 14) rated their preference for 72 individual food items, each of these mapping to one of six categories encompassing high and low-fat choices in combination with sugar, complex carbohydrate or and protein. An 18-item buffet meal was created for each participant based on expressed preferences. Overall energy intake was reduced on average by 60% at the 24-month buffet meal. Reductions in intake were seen across all six food categories. Decreases in the overall intake of all individual macronutrient groups were marked and were generally proportional to reductions in total caloric intake. Patterns of preference and intake, both at baseline and at follow-up appear more idiosyncratic than has been previously suggested by verbal reporting. The data emphasise the consistency with which reductions in *ad libitum* food intake occur as a sequel of RYGB, this being maintained in the setting of a self-selected *ad libitum* buffet meal. Exploratory analysis of the data also supports prior reports of a possible relative increase in the proportional intake of protein after RYGB.

## 1. Introduction

Roux-en-Y Gastric Bypass (RYGB), is one of the most commonly used surgical interventions worldwide for obesity, resulting in 20–30% reductions in body weight within the first post-operative year in the majority of patients [1]. The gradient of weight loss following surgery is not constant; being characterized by a steep period of accelerated weight loss during the first 6 post-operative months tapering thereafter towards weight stability within 18–24 months [2,3,4]. Although in the longer term, many patients experience some degree of weight regain, weight typically remains substantially reduced relative to pre-operative level [5,6]. In this sense, the trajectory of weight loss, prolonged weight-loss maintenance, and attenuated weight regain observed after bariatric surgery stands in contrast to the common finding of weight-loss recidivism within 2 years of follow-up after diet and lifestyle interventions [7,8]. 

Reductions in energy intake are observed following RYGB, particularly during the first post-operative year [9,10,11,12,13]. Over time following RYGB, whilst energy intake has been reported to increase, it does not typically return to pre-operative levels [12,13,14]. 

Uncertainty exists regarding the key mechanisms underpinning reductions in energy intake after RYGB. The relative importance of decreased overall portion sizes versus shifts in food preference and intake away from energy-dense foods remains an area of active investigation [15]. Studies in rat models of RYGB demonstrate a shift in preference away from high fat and high sugar foods after surgery [16,17,18,19]. In human clinical research, several studies report that after bariatric surgery, patients shift their preferences from high fat and high sugar foods to healthier options such as fruits and vegetables [9,10,16,20,21,22]. The studies reporting these changes have relied on verbal reporting and food questionnaires [23,24,25]. 

Various hypotheses have been advanced to explain why food preferences might change post-surgery. Patients do often report post-prandial anxiety and nausea, when foods high in simple sugars are ingested, consistent with early dumping syndrome. This could in turn result in conditioned avoidance of certain foods [26,27,28]. Another potential mechanism involves alterations in taste sensitivity for sugary foods as well as reduced palatability of high fat foods [19]. Reductions in the hedonic value of energy-dense foods could also constitute a reason for a shift to healthier options [26,27,29].

Recently Neilsen et al., used direct measurement of food intake at an investigator defined *ad libitum* buffet meal before and after surgery [30]. The authors did not observe changes in food preference in patients undergoing RYGB or sleeve gastrectomy (SG) [30] but did report that the magnitude of decrease in energy intake, the selection of less energy dense foods, and increases in protein intake found at 6 months after surgery, predicted the subsequent level of weight loss achieved at 18-month follow-up [31]. 

The approach used in the present study sought to further validate the impact of RYGB on food intake by asking study participants to consume *ad libitum* from an 18-item buffet meal, the components of which were personalized to each participant based on their own ratings at baseline of 72 individual food items across 6 categories using a validated food preference questionnaire (FPQ) [32]. Changes in overall food intake were assessed between buffet meal study visits conducted at pre-operative baseline, and at 24 month follow up after surgery. The percentage contribution to energy intake according to macronutrient group was explored in parallel. We hypothesised that directly measured reductions in food intake after RYGB surgery would persist in the context of a self-selected, personal preference based, ad libitum buffet. Through detailed recording of the food selected, we sought to investigate whether the intake of any food category or macronutrient group was consistently altered as a component of overall reductions in intake. 

## 2. Materials and Methods

### 2.1. Study Population

The study was approved by the Ethics and Medical Research Facility, St. Vincent’s University Hospital. Participants referred to the Bariatric Surgery Service at St Vincent’s University Hospital, St Columcille’s Hospital, Loughlinstown, Letterkenny General Hospital and The Cork Clinic, Ireland were considered for recruitment between June 2016 and July 2019. Inclusion criteria for the study stipulated that to be eligible as participants, study volunteers were to be scheduled to undergo RYGB, be independently mobile, have capacity to consent to participate and be at least 18 years of age. Exclusion criteria included any systemic or gastrointestinal condition which may affect food intake or preference, pregnancy or breast feeding, medications with a documented effect on food intake or food preference, a history of significant food allergy, any specific dietary restrictions or any significant cognitive or communication issues. 

RYGB surgery involves the division of the stomach into two parts—an upper stomach pouch and a lower remnant stomach which remains in situ in continuity with the duodenum. The stomach pouch (15–30 mL) is anastomosed to the distal limb of a mid-jejunal transection, created 30 to 75 cm distal to the ligament of Treitz (Roux limb). In order to preserve continuity of the intestine, jejunojejunostomy is then completed by anastomosis of the proximal limb of the jejunotomy to a point 75–150 cm distal to the gastrojejunostomy. Therefore, food transits straight to the small intestine and the biliopancreatic secretions and secretions from the remnant stomach meet with the ingestants in the common channel, which runs distal to the Y-shaped jejunojejunostomy [33].

Informed written consent was obtained from 26 eligible participants and 14 (63%) conducted both baseline and 24-month follow-up study visits We present data drawn from repeated measures analysis of these 14 study completers, formally comparing total and macronutrient specific intake across study visits and exploring underlying patterns of food preference and evidence of any consistent changes therein.

### 2.2. Food Preference Questionnaire

At the screening visit participants were asked to rate their preferences for 72 different food items from a 72-item FPQ based on the model developed by Geiselman et al., incorporating high and low-fat content items in complex with protein, simple sugar or complex carbohydrates [32]. We refined the questionnaire to suit the Irish population by including common food items widely available in Irish supermarket stores. Items categorised into one of six different groups depending on their macronutrient character were listed in no particular order. Participants were asked to rate each individual item along a 9-point scale with the following anchors: 1 = dislike extremely; 5 = neutral; 9 = like extremely. Responses from the FPQ were used to select appropriate food items from the stock list of 72 items (Appendix A) for use in a personalised 18-item buffet spread. For each participant the total overall score recorded for the FPQ food preference questionnaire was calculated out of 648 (9 highest Likert scale ranking ×72 items of food). 

### 2.3. Study Visits 

The baseline study visit was conducted one month before surgery and the comparator study visit took place 24 months after surgery. Variable numbers of participants also attended for study visits at 3- and 12-month follow-up. For each visit the participants arrived at 1030 at the St Vincent’s University Hospital Clinical Research Centre (CRC). All participants were fasting from 2200 the previous evening. Weight, height and waist circumference were recorded. At 1100, a room in the CRC was prepared with an 18-item buffet spread selected based on their FPQ responses at screening. High fat and low-fat options were combined with varying simple sugar, complex carbohydrate and protein content, as depicted in Table 1. The study endeavoured to provide each participant with foods to which he/she had assigned an intermediate hedonic rating of between 5 and 7. High fat foods given to an individual participant were paired with a low-fat variation of the same food, e.g., high- and low-fat meat options, high- and low-fat cheese options. The order in which foods were presented in the buffet spread was randomised at each study visit. Containers of known weight of the 18 test foods were provided to each participant. In addition to this, basic condiments such as mayonnaise, ketchup, peanut butter, butter, spread, strawberry jam, salt, pepper along with drinks such as coke, diet coke, sprite, diet sprite in cans and carton milk were also provided. Two 500 mL bottles of water were also supplied.

Participants were instructed to eat their meal within 60 min and consume as much food as they wished in order to feel “comfortably full”. They were asked to switch off their electronic devices. After 60 min, once the participant had left a member of the research team recorded the exact quantity of each food item chosen by each participant by determining change in test food container weight. The same researcher used the same weighing scale throughout the study to weigh every item. The amount of macronutrient consumed from each food item was recorded and its caloric value calculated based on nutritional information labelling provided by the producer. Percentage contributions of fat, total carbohydrate, simple sugar and protein to overall energy intake were calculated.

### 2.4. Experimental Power and Inferential Statistics 

The study aimed to satisfy requirements for the detection of a reduction of one-third in *ad libitum* total food intake as the primary outcome, controlling for type 1 and type 2 error rates of 5% and 20% respectively and recruiting at approximately 200% power to mitigate for potential drop out between the baseline visit and 24 months.

Data assessed by inferential statistics were expressed as mean ± standard error of the mean (SEM). Paired t-tests (parametric data-weight, waist circumference and absolute caloric intake) were used to compare baseline vs 24 months data. A two-sided Fisher’s exact test was carried out to compare proportions of participants who did and did not consume a specific food category at baseline vs at 24 months. Results were deemed significant at when *p* < 0.05. 

The data pertaining to preference across macronutrient groups are exploratory in nature and presented as mean difference, standard deviation and associated 95% confidence interval. All data were analysed using GraphPad Prism^®^ software (Version 4) (GraphPad Software, Inc., San Diego, CA, USA).

## 3. Results

### 3.1. Study Participants 

A total of 14 participants were included in the analysis. Table 2 shows the baseline characteristics of these participants. The study cohort was 70% female with a mean age was 51.0 ± 7.8 years. Mean BMI was 50.1 ± 7.8 kg/m^2^ at baseline.

### 3.2. Surgery Induced Changes in BMI

At 24 months post-RYGB, BMI had decreased from 50.1 ± 7.8 kg/m^2^ to 35.9 ± 7.8 kg/m^2^ (*p <* 0.01) (Figure 1). Waist circumference reduced from 140.5 ± 14.7cm to 114.6 ± 18.4 cm (*p <* 0.01).

### 3.3. Questionnaire Based Food Preference Ratings at Baseline 

Totals from the pre-operative food preference rating questionnaire showed a range of overall ratings for the 72 items extending from a minimum rating of 321 (49.5%), to a median rating of 406 (62.7%) and a maximum rating of 514 (79.3%) (Figure 2). The Median ratings for the six food categories were: High Fat/ High Sugar—5.75, High Fat/High Complex Carbohydrates—6.5, High Fat/High Protein—5.25, Low Fat/High Sugar—7, Low Fat/High Complex Carbohydrates—7 and Low Fat/High Protein—6.25. The heat map (Figure 2) shows the ratings on the Likert scale and the selections made from the FPQ for each food category for the participant with minimum, median, and maximum rating.

### 3.4. Changes in Absolute Intake 

At 24 months post-surgery, food intake was on average reduced by 60.9% with respect to baseline intake (*p* < 0.01) (Table 3), exceeding by approximately two-fold conservative *a priori* estimates of an attenuated reduction in food intake when food items are presented based on self-selection. With respect to individual macronutrients, there was a 59% reduction in fat intake from baseline to 24 months post-surgery (*p* < 0.01), a 67% reduction in sugar intake from baseline to 24 months post-surgery (*p* < 0.01), a 67% reduction in total carbohydrate intake from baseline to 24 months post-surgery (*p* < 0.01) and a 43% reduction in protein intake from baseline to 24 months post-surgery (*p* < 0.01) (Table 3). 

Intake from the 6 food categories varied; and not all categories were consumed by all participants at baseline and 24 months post-surgery (Figure 3). As a general pattern, consumption of all food categories was reduced at follow-up. At baseline, 57% consumed High Fat /High Sugar items and this dropped to 28.5% of participants at follow-up (*p* < 0.01). High Fat /High Complex Carbohydrates items were consumed by 71% of participants at baseline and by 41% of participants at follow-up. At baseline 85% consumed High Fat/High Protein items, this dropped by 33% to 57% of participants at follow-up (*p* = 0.03). Low Fat/High Sugar items were consumed by 85% at baseline and by 57% of participants at follow-up. Low Fat/High Complex Carbohydrate items were consumed by 85% baseline, and this dropped by 25% to 64% of participants at follow-up (*p* = 0.01). Low Fat/High Protein items were consumed by 78% of participants at baseline and by 57% of participants at follow-up (Figure 3).

No consistent trends were observable with respect to changes in the proportional representation of fat, simple sugars or total carbohydrates in food consumption patterns across visits (Table 4). However, the data did demonstrate a consistent trend towards an increased representation of protein as a proportion of overall consumption. Based on the wide-confidence interval, the computed power for detection of this magnitude of effect in the present data set was 75%, and hence we report the result in Table 4 as a trend. However, interrogation of the source of the width of the confidence interval reveals that one particular participant consumed only rice cakes, turkey slices and 0% fat-high protein yoghurt at their 24-month visit, for a total energy intake 895 kJ, this being the lowest among all participants at 24 month follow-up. This consumption pattern corresponds to a protein preference of 73%. The group average protein preference at 24 months when this participant’s data are removed is 20.8% (S.D. 10.8), resulting in a mean difference of +6.8% and a narrowing of the 95% confidence interval to 0.34 to 13.2. This sub-analysis meets 80% power and is significant at a *p*-value of 0.04 on paired analysis.

## 4. Discussion

Verbal reporting tools are inherently subjective and can be prone to recall bias in terms of under reporting meal size of foods perceived as unhealthy [23,24,25,34] especially in people with obesity in which there may be issues of stigma [35]. The results from this study are not in complete disagreement with prior findings based on verbal report [9,11,12,16,20,21,36,37,38,39,40,41,42,43], but by using a direct measurement-based approach [30,31] eliminate inaccurate recall as a potential source of error.

An important distinction of the present study relative to recent reports is that by factoring in individual patterns of food preference at pre-operative baseline and using this to inform the creation of personalized buffets, the study was equipped to reveal whether the post-surgical reductions in intake previously described persist in a setting wherein participants are invited to consume *ad libitum* from an individual preference determined spread of food items. We asked participants to- “eat as much as they like till comfortably full” which proved successful with more than 78% of participants managing to eat over 5000 kJ at baseline, with the highest amount of energy intake consumed being 13,096 kJ. 

Whilst the FPQ approach did point to a relatively higher hedonic rating in general for foods containing sugar, there was real diversity in the nature and breadth of expressed preferences. This highlights the likely multifactorial genesis of food preference in humans which may defy its classification according to a reductionist perspective on the relationship between taste, and appetitive and consummatory behaviour. The overall reduction in the quantity of food consumed across all macronutrient categories at 24-month follow-up was the most striking feature of the analysis. It may be argued that participants ate less in order to meet with their perception of what researcher expectations would be of how they should behave after surgery. This cannot be completely discounted but were it the case, a clearer shift in apparent preferences towards calorie sparse and stereotypically healthy choices might have been anticipated There was in fact a reduction in the proportion of participants choosing food items from across the six defined categories, when buffet intake was assessed by macronutrient type, there was no clear evidence of reduced preference for high fat, high sugar items as had been hypothesized. There was, however, some evidence that after RYGB, there is an increase in the percentage contribution of protein to energy intake. The large magnitude reduction in total *ad libitum* intake is in agreement with the recent findings from researcher defined buffet meals [30,44,45]. None of these studies resolved a proportional change in food preference, but it could be argued that none took into account baseline individualised preferences. It is interesting however to note that increases in the relative contribution of protein to overall intake at a buffet meal 6 months after surgery showed a modest correlation with weight loss at 18 months [31]. This observation in combination with the indication from the present study of a shift towards low-fat, protein enriched food item selection could be suggestive of a link between lean mass loss as a component of total weight loss after surgery and an increase in preference for high protein items in the diet. This concept has not been widely discussed as a potential driver of changes in dietary choices after surgery, which has hitherto focused more on palatability, reward and post-ingestive symptoms as an explanation. In rats, dietary protein restriction conditions animals towards protein seeking behaviour [45]. By analogy, it is tempting to speculate that lean mass loss—an undesirable but well acknowledged consequence of surgery amenable to treatment with amino-acid supplementation [46]—could lead patients to preferentially select protein rich food items as part of a process of physiological compensation after surgery. 

The results of the present study differ from findings obtained using a comparable paradigm in preclinical research in rats, Mathes et al., used a cafeteria diet incorporating low fat and high fat options coupled with low sugar and high sugar. The study reported a shift away from fat towards an increase in carbohydrate intake after RYGB which was not seen here. It is worth noting, however, that Mathes et al. [19] also reported a slight but significant increase in relative protein intake after RYGB in rats. However, apparent discrepancy between the findings from animal and human studies employing direct measures of choice and intake may be more apparent than real. Rodents are not subject to nutritional counselling. Moreover, the same cognitive, economic, and social factors that are part and parcel of human feeding are not at play in animal models. Finally, whereas rodents are raised on a single diet, sometimes a choice of two, humans have a lifetime of diverse experiences with a variety of foods. Given that there is evidence in rodent models that learning may play a role in guiding food choices after RYGB [16,18,19,47], the relative lack of experience with food variety might better position them to associate flavour cues with post-ingestive events to subsequently guide their ingestive behaviour. Thus, it is important not to dismiss the translational relevance of the animal model studies as they might reflect the most fundamental processes engaged by RYGB to alter food selection and relative macronutrient intake when other influences specifically affecting human ingestive behavior are stripped away. 

In recognizing the limitations of the study, the potential for selection bias arising from sub-optimal retention rate must be acknowledged. All participants had achieved and sustained a significant level of weight loss and hence the patterns of change described in the data are restricted in their relevance to the context of patients achieving a good clinical response to surgery. Unfortunately, we do not have a record of trajectory of body weight in those participants not attending for 24-month study visit. It cannot be definitively excluded that a sub-optimal response to surgery with respect to weight-loss trajectory did not have a bearing upon likelihood of attending. The ad libitum buffet meal, although a novel assessment technique, also has some limitations. We cannot ignore the fact that eating in a clinical research centre is not reflective of actual behavior at home and the power of assessing consumption at a single meal as means of describing patterns of change in eating behaviour is potentially susceptible to day-to-day variation. 

Residential studies may mimic more free-living conditions and allow for more sampling and provide more precise estimates of intake patterns. The results of these types of studies are awaited [48]. The ad libitum buffet meal was picked from the 72-items list and a researcher chose 18 items from this list. The items were chosen between 5 and 7 liking on a Likert scale and although all efforts were made there may be subjective researcher bias when selecting these items. During screening there were individuals who ranked all items between 5 and 7, while other subjects ranked all items between 1 and 4 or 7 and 9. This made the decision harder and, therefore, could influence the actual liking and dislike of foods and the 3 items per group that were chosen. Video recording of the buffet would permit reporting of parameters such as meal duration and rate of eating, factors we were unable to report on from our study and the opportunity to assess more comprehensive microstructural features of ingestive behaviour [48]. 

## 5. Conclusions

Using direct measurements of ad libitum intake during a buffet meal, this study confirmed that the major reductions in ad libitum food intake seen earlier after surgery are sustained out to 24 months after RYGB. Deselection of high fat and/or high sugar food items was prevalent but not universal, suggesting that after RYGB there may be subgroups of patients differing in their adaptive response to surgery with respect to food selection. Although absolute intake of all macronutrients declined, there was evidence of of a small increase in the proportional intake of protein. Whether this observation can be corroborated in larger more granular studies and whether it truly reflects a change in subjective preference remains an interesting question. 

## Figures and Tables

**Figure 1 nutrients-13-03873-f001:**
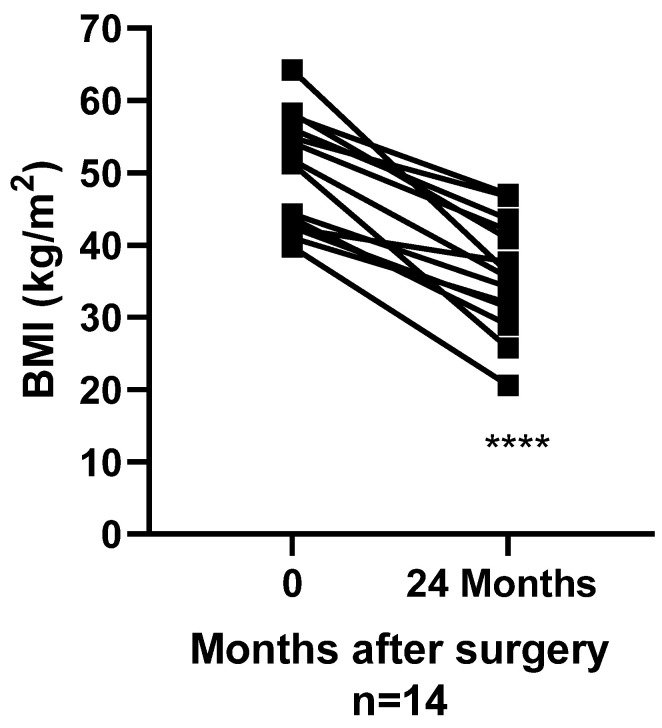
Body weight change 24 months post-surgery. Change in body weight represented by BMI from baseline to 24 months post-surgery. *n* = 14, ****: *p* < 0.0001, BMI Body Mass Index.

**Figure 2 nutrients-13-03873-f002:**
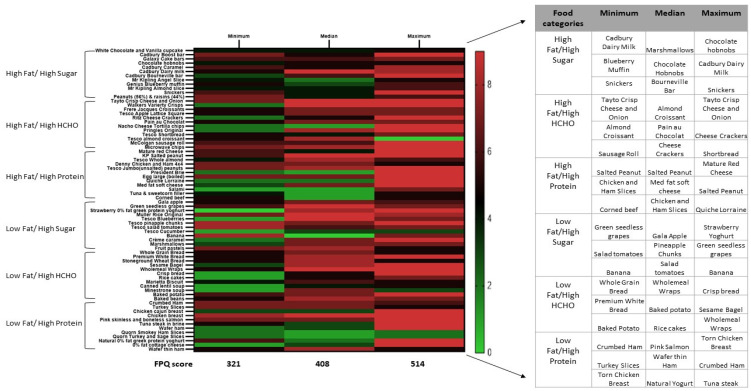
Questionnaire-derived food preference ratings and meal selection. HCHO, High Complex Carbohydrates; FPQ, Food preference questionnaire. The heatmap shows graded preference of the 72 items for three participants (minimum, median and maximum ratings). The colour intensities vary from green to red where green is a score between 0 and 3 and red between 6 and 9 on the Likert scale.

**Figure 3 nutrients-13-03873-f003:**
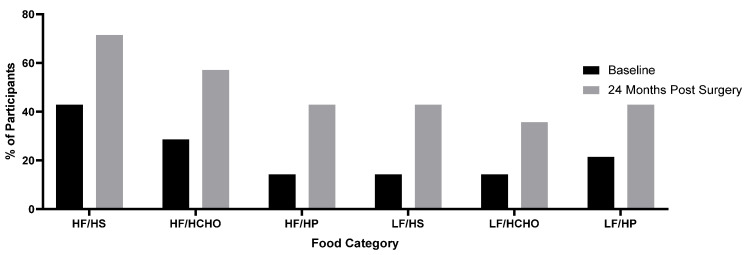
Proportion of Participants Not Consuming Items from Each Category, baseline versus 24 months. Percentage of participants not consuming each category at baseline and at 24 months post-surgery. HF/HS = High Fat/ High Sugar, HF/HCHO = High Fat/High Complex Carbohydrates, HF/HP = High Fat/High Protein, LF/HS = Low Fat/High Sugar, LF/HCHO = Low Fat/High Complex Carbohydrates, LF/HP = Low Fat/High Protein.

**Table 1 nutrients-13-03873-t001:** Macronutrient self-selection paradigm.

Macronutrient Self-Selection Paradigm			
	High Simple Sugar	High Complex CHO	Low CHO/High Protein
High Fat	Three foods	Three foods	Three foods
	Fat > 40%	Fat > 40%	Fat > 40%
	Sugar > 30%	Comp Carb > 30%	Protein > 13%
Low Fat	Three foods	Three foods	Three foods
	Fat < 20%	Fat < 20%	Fat < 20%
	Sugar > 30%	Comp Carb > 30%	Protein > 13%

Adapted from Geiselman et al. [32].

**Table 2 nutrients-13-03873-t002:** Baseline characteristics of participants (*n* = 14).

Parameter	Values
Male/Female	4/10
Age (years)	51.0 ± 7.8
Body weight (kg)	136 ± 23.8
BMI (kg/m^2^)	50.1 ± 7.8
Waist Circumference (cm)	140.5 ± 14.7

Data presented as mean ± standard deviation. BMI: Body Mass Index.

**Table 3 nutrients-13-03873-t003:** Absolute measures of total and macronutrient specific intake at buffet meal. Data presented as mean ± standard error.

	Pre-Surgery	24 Months Post-Surgery	Mean Difference (95% CI)	*p*-Value
Total food intake (kJ)	6510.3 ± 756.5	2540.1 ± 431.0	−3971.8 (−5234.2 to −2707.1)	<0.01
Intake from macronutrients				
Fat (kJ)	2617.1 ± 318.8	1066.9 ± 225.5	−1548.1 (−524.8 to −216.1)	<0.01
Sugar (kJ)	1312.5 ± 288.3	421.7 ± 115.9	−890.8 (2195.7 to −334.5)	<0.01
Total Carbohydrates (kJ)	3102.4 ± 455.6	1013.8 ± 220.9	−2088.7 (−2888.2 to −1289.1)	<0.01
Protein (kJ)	833.9 ± 87.0	472.8 ± 58.2	−558.1 (−558.1 to −163.2)	<0.01

**Table 4 nutrients-13-03873-t004:** Percentage contribution of individual macronutrients to energy intake at buffet meal Data presented as mean ± standard deviation.

	Pre-Surgery	24 Months Post-Surgery	Mean Difference	95% CI
Fat (%)	40.6 ± 11.6	38.3 ± 18.1	−2.26	−14.1 to 9.5
Sugar (%)	18.7 ± 8.7	15.5 ± 12.1	−3.15	−13.1 to 6.8
Total Carbohydrates (%)	46.4 ± 10.5	38.4 ± 17.2	−8.02	−20.9 to 4.9
Protein (%)	13.4 ± 3.7	24.1 ± 17.4	+10.72	0.3 to 21.1

## Data Availability

The datasets used and/or analysed during the current study are available from the corresponding author on reasonable request.

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
