# Peer review of "A Comparison of Total Food Intake at a Personalised Buffet in People with Obesity, before and 24 Months after Roux-en-Y-Gastric Bypass Surgery"

_nutrients, 2021, doi:10.3390/nu13113873_

Round 1

Reviewer 1 Report

The concept of the study is interesting! However, the small number of inclusions limits the validity of the conclusions. I invite the authors to increase the number of participants as well as the duration of follow-up.

Reviewer 2 Report

In this Article the authors followed on previous observation that gastric bypass reduces food intake and they hypothesized that this reduction is mostly due to lower intake of high dense foods such as High fat diets. They did observe in fact a uniform reduction in total caloric consumption 2 years post RYGB. This  confirms previous observation related to this finding.

The interesting part of their study is that they planned a good control for food selection pre- and post surgery , taking into consideration combination of fat and sugar as a driver for palatability and food preference in many people with food addiction. This approach and design in planning , increased the value of their findings , specifically that related to decrease in food choice selection among participants post-RYGB!

the second interesting finding is related to the relative increase in % of protein intake relative to total caloric  consumption. Their study is well designed and the data are nicely dsipalyed.

I have few comments tough:

1- the authors mentioned in the discussion  line 64, that " reductions in the quantities of food consumed after RYGB have primacy in terms of magnitude and consistency over changes  in food selection indicative of alterations in preference , which tend to be idiosyncratic". I do not believe that the authors can make such an assumption or conclusion based on a on observation from 14 individuals over 2 years ?!?

2- it would be helpful if the authors have data / scores of other hedonic measures of food intake to include pre- or post RYGB. 

Round 2

Reviewer 1 Report

I think that despite the explanations of the authors, the power of the study remains very low to extrapolate and validate the results

Author Response

We accept the reviewer's point that the number of completers in the study does limit the power of the study to detect subtle changes in preference. We have further acknowledged this in the discussion and tempered further our conclusions therein. Whilst there may be subtle but important differences in preference which lay beyond the statistical power of the present study, they would appear to be dominated by quantitative reductions in food intake, which can with reasonable confidence be said to occur across all macronutrient groups. The small signal of an increase in relative preference for protein does warrrant better powered examination, perhaps in a more free-living setting with multiple direct measures of intake to allow for a consistent pattern to emerge more strongly. This point has been made in the discussion.